# Design and Development of a Broadband Vibration Energy Harvester Suitable for Tractor Exhaust Cylinder Vibration

**DOI:** 10.3390/s23010286

**Published:** 2022-12-27

**Authors:** Xinxin Ma, Tianshuo Zhou, Lijiao Gong, Xin Zhang, Fuyuan Yao, Chujian Wang

**Affiliations:** 1School of Mechanical and Electrical Engineering, Shihezi University, Shihezi 832000, China; 2Key Laboratory of Northwest Agricultural Equipment, Ministry of Agriculture and Rural Areas, Shihezi 832000, China

**Keywords:** vibration energy, piezoelectric energy harvester, broadband, tractor, sensor

## Abstract

A large amount of vibration energy exists in the working environment of tractors. Therefore, the use of vibration energy harvesting technology to convert the vibration energy into electrical energy is a feasible way to supply power to low-power sensor equipment in agricultural machinery. Aiming at the problem in which the internal sensors of traditional tractors require built-in batteries or overlapping cables, this work proposes a broadband piezoelectric vibration energy harvester that could harvest the vibration energy from the tractor exhaust cylinder when the tractor is working. The vibration energy can be converted into electrical energy to power the air pressure sensor device. This experimental investigation shows that the energy harvester is composed of a folded piezoelectric energy harvester and a multi-source input synchronous electronic charge extraction circuit.The circuit has a high power density of 12,398 μW/(mm^3^·g^2^). Hence, it can convert vibration energy into a wide frequency range between 90–140 Hz and cause the air pressure sensor to operate.

## 1. Introduction

Tractors, as large agricultural machinery equipment, inevitably produce considerable vibration in their own operation and working environment. With the development of science and technology, an increasing number of micro-power sensor devices are used to monitor and collect data in agricultural production in real time [1]. At the same time, traditional energy problems have become increasingly prominent. As a new energy source, vibration energy harvesting technology has also received widespread attention. It provides a solution for the power supply of equipment in agricultural Internet of Things technology. The harvest and utilization of vibration energy in production and living environments cannot only reduce traditional energy consumption and emissions but also suppress the damage to the structure caused by vibration [2]. The development of tractors in the direction of intelligence and automatic driving is inseparable from the intervention of numerous sensor devices, and the resulting power supply problems for the equipment are prominent. Due to the differences in the total mass, size, chassis geometry, suspension characteristics and driving environment, tractors have a higher level of vibration intensity [3]. Therefore, harvesting and using the vibration energy generated during tractor operation not only solves the power supply problem of the sensor equipment in the tractor but also plays a role in reducing vibration and noise, which is of great significance to promote the development of intelligent tractors. Fan and others studied that the position of the exhaust cylinder and the vibration intensity of the driver’s seat floor will increase sharply with the static speed of the Dongfanghong-354 tractor. The growth rate is significantly higher than other areas of the devices [4]. Vibration energy harvesting technology can be piezoelectric, electromagnetic and electrostatic. Piezoelectric has been widely studied and has numerous advantages, including anti-electromagnetic interference, easy integration and a simple structure [5,6,7,8].

Many researchers have studied the vibration performance and vibration energy utilization of vehicles. Mohamed et al. [9] analyzed and discussed the average power and total recoverable power of the vehicle damper under two different load conditions, namely, full-load and no-load, and under different driving environments of heavy vehicles. The parameter bandwidth sensitivity analysis of the vehicle recoverable damping power and suspension dynamics was carried out, which provided the basis for the design of the broadband energy harvester and could improve the energy acquisition efficiency under different driving road conditions. Li Peng [10] of Huazhong Agricultural University converted the vibration energy generated by the diesel engine of agricultural vehicles into electrical energy and stored it in a super capacitor through experiments to charge the rechargeable battery further. Majid Khazaee et al. [11] combined a cantilever beam piezoelectric harvester with a pump to use the vibration energy during operation to power a pump operating condition monitoring device for autonomous fault diagnosis. Jingui Qian et al. [12]. proposed an on-vehicle magnetically triboelectric nanogenerator (V-TENG) to harvest the rotational energy of tires. The 100 rpm V-TENG device can provide a maximum output voltage of 316 V and a power of 22.3 mW at a load of 3 mΩ, thereby providing direct power to the tire pressure monitoring systems and wireless sensor nodes. However, there are still the problems of an inefficient energy transfer and low reliability. Although multi-modal harvesters have been reported in literature (U-shape [13], M-shape [14], H-shape [15],trapezoidal [16], etc.), generating relatively large power outputs around each mode and effectively widening the bandwidth using the coupled multi-modal structures is a difficult task. In many studies, the harvester had multiple modes but the resonant frequencies were not close enough to each other, or only one vibration mode was dominant. This significantly affected the energy transfer efficiency of the piezoelectric energy harvesting system.

In this paper, the authors proposed a broadband piezoelectric vibration energy harvester that could harvest the vibration energy from the tractor exhaust cylinder when the tractor is operating. The vibration energy was converted into electrical energy to power the air pressure sensor device. The energy harvester consisted of a folded piezoelectric energy trap with a resonant frequency close to the MSI-SECE circuit with a high power density of 12.398 μW/(mm^3^·g^2^). This paper explored the vibration of a John Deere-454 exhaust cylinder and used piezoelectric vibration energy harvesting technology to harvest the energy generated by vibration, which drove the pressure sensor for the tractor exhaust cylinder to work normally in a wide frequency range. Section 2 describes the structure of the energy harvester system based on the vibration of the tractor exhaust cylinder through the test and analysis of the vibration source, the structure of the piezoelectric energy harvester and the energy acquisition circuit. Section 3 builds an experimental platform for the circuit and power supply tests. Section 4 includes the discussion and conclusion of this study.

## 2. System Principle and Design

Vibration exists widely in the operation of tractors. If it can be harvested and used, then it can also reduce vibration and noise. A piezoelectric power supply system consists of a vibration source, a piezoelectric structure, an energy harvesting circuit, an energy management circuit and load (Figure 1). A broadband vibration energy harvester suitable for the vibration of the tractor exhaust cylinder is designed on the basis of its test and analysis. The structural size of the folded oscillator is determined by the modal analysis. The charge generated by the upper and lower surfaces of the piezoelectric plate is stored in the energy storage element through the energy harvesting circuit. The DC–DC management circuit is used for the pressure sensor of the tractor exhaust cylinder.

### 2.1. Vibration Source Test and Analysis

In this paper, we propose the combination of vibration energy harvesting technology with the tractor. On the one hand, this technology harvests vibration energy to supply power to the sensor equipment. On the other hand, it can reduce vibration and noise. First, the vibration test experiment was carried out to determine the vibration source, that is, the tractor exhaust cylinder. In the Northwest Agricultural Equipment Key Laboratory of the Ministry of Agriculture and Rural Affairs, the vibration test of the tractor exhaust cylinder at five different positions was completed when the John Deere-454 tractor engine speed was maintained at 650, 800, 1100, 1300 and 1500 r/min.

#### 2.1.1. Test Platform Construction

The John Deere-454 tractor exhaust cylinder was taken as the test object, and five measuring points (No. 1–5) were eventually selected on the exhaust cylinder. The positions of the five measuring points are shown in Table 1. The schematic of the test experiment platform is shown in Figure 2a. The five three-axis acceleration sensors were steadily adsorbed into the corresponding measurement points of the exhaust cylinder through the magnetic suction seat in the sequence. The sensor output was connected to the data acquisition instrument. The data were recorded and uploaded to the host computer in real time.

The test platform is shown in Figure 2b. The direction of the tractor head was the X-axis. The direction horizontal and vertical to the *X*-axis was the *Y*-axis. The direction perpendicular to the ground was the *Z*-axis. The tractor acceleration sensor, which can measure the *X*, *Y* and *Z* axes simultaneously, was obtained from the Beijing Oriental Institute of Vibration and Noise Technology (hereinafter referred to as the Beijing Oriental Institute). This tractor acceleration sensor has a built-in IEPE pre-amplified three-axis acceleration sensor INV9832-50. The sensitivity was 100 mV/g and the frequency range was 0.4–12,000 Hz, which met the needs of the vibration test of the tractor exhaust cylinder.

The data acquisition instrument was selected for the network distributed acquisition instrument INV3062S produced by Beijing Dongfang. It is a high-precision data acquisition test suitable for structural vibration, noise, strain and other physical signals. It is windproof and adapts to the harsh on-site working environment. The technical indicators of the data acquisition instrument are shown in Table 2.

#### 2.1.2. Analysis of the Test Results

The test results above were processed by the signal analysis software (Coinv DASP V11) (Figure 3), and the vibration acceleration of the different parts of the exhaust cylinder were obtained from the time domain analysis of the acceleration waveform of the three directions at the five measurement points of the exhaust cylinder.

The vibration acceleration of most of the measurement points of the exhaust cylinder increased with the tractor engine speed. The No. 1 measurement point was in the Z direction, which means that the acceleration in the vertical direction was the largest. The acceleration in the three directions of the No. 2 and No. 4 measurement points were maintained at about 2 m/s^2^. The acceleration of the No. 3 and No. 5 measuring points in the X direction were the largest. The effective value of the acceleration of each part of the exhaust cylinder was within the range of 0.39–3.3 m/s^2^. In the directions X and Y, the acceleration on the left side of the exhaust cylinder muffler changed greatly at different speeds. In the Z direction, the accelerations on the right and left sides of the exhaust cylinder muffler changed slightly.

A power spectral density (PSD) analysis was performed on the measured data at the five measuring points in Table 1 and at the constant speed of 650, 800, 1100, 1300 and 1500 r/min. As shown in Figure 4a, in the X direction at 650–1500 r/min rotational speed, the PSD of the five measuring points within 700 Hz was approximately 0.1 g^2^/Hz and the vibration level was high. The PSD of the No.4 measuring point was higher in the range of 500 Hz. At 800 r/min, the PSD of No. 5 measuring point was 0.4–0.5 g^2^/Hz. As shown in Figure 4b, in the Y direction, the No. 4 measuring point was within the vibration frequency of 70–120 Hz, and the PSD reached 0.5–0.6 g^2^/Hz. The PSD of the No. 2 measuring point was relatively high in the range of 0–500 Hz, which was approximately 0.2 g^2^/Hz. In the frequency range of 20–390 Hz, the overall vibration level of the five measuring points was high. As shown in Figure 4c, in the Z direction, the PSD fluctuated, staying within 0.1 g^2^/Hz as the speed increased. In the frequency range of 0–500 Hz, the PSD of the No. 2 measuring point was at a relatively high level as the speed increased. However, in the range of 600–1000 Hz, the PSD exhibited a significant downward trend.

In summary, the overall vibration of the tractor exhaust cylinder was obtained according to the test data analysis. When idling at 650–1500 r/min, the vibration frequency of the exhaust cylinder itself was within 1000 Hz and the acceleration was 0.39–3.3 m/s^2^. According to the PSD analysis, the PSD of the X, Y and Z directions of the five measurement points were relatively large within the frequency of 600 Hz. The overall vibration of the exhaust cylinder was relatively severe, especially in the frequency range of 20–400 Hz. Therefore, to harvest the vibration energy of the tractor exhaust cylinder in the idling state, the piezoelectric energy harvester should have a better frequency response in this vibration frequency band.

### 2.2. Piezoelectric Energy Harvester with Folded Structure

The team previously proposed a multi-mode piezoelectric energy harvester—a folded structure piezoelectric energy harvester (hereafter referred to as folded energy harvester). By combining the vibration testing and analysis of the tractor exhausts in Section 2.1, the structural dimensions of the folded piezoelectric transducer must be fully considered for the vibration energy harvesting applications so that it can have a better wide-frequency response in the range of 20–400 Hz.

The modal simulation analysis of the folded energy harvester was carried out by the ANSYS software (Figure 5a). The final size of the piezoelectric energy harvester in this application was developed and determined to be 34 mm × 13.6 mm × 18.4 mm, with a piezoelectric plate size of 31 mm × 13.6 mm × 0.22 mm, a gap width of 2 mm, a mass block thickness of 4 mm and a thin middle beam thickness of 1 mm. The first two orders of the vibration mode is shown in Figure 5a. The first-order and second-order bending vibrations were determined. Also, a more uniform strain can be generated in the middle thin beam. The frequency response in the range of 90–140 Hz was good.

The folded structure piezoelectric energy harvester is shown in Figure 5b. The main structural material is stainless steel (304). Lead zirconate titanate (PZT) is also a widely used piezoelectric material. The structural parameters are shown in Table 3. The piezoelectric materials bonded to the elastic beam in the middle are the main parts of the vibration energy conversion that affect the vibration energy conversion efficiency directly [17]. PZT was affixed to the upper and lower surfaces of the elastic thin beam layer of the folded energy harvester. After dividing the electrode surface, the four outputs were obtained by setting four leads, namely, V*p*_1_, V*p*_2_, V*p*_3_ and V*p*_4_. The core components of the folded energy harvester are shown in Figure 5b.

The four outputs obtained after the division of the electrode surface on PZT, which is based on the elastic beam in the middle of the folded piezoelectric energy harvester, had different phase relationships. These outputs had a direct impact on the energy extraction strategy and efficiency. The vibration platform was maintained at a constant acceleration of 0.5 m/s^2^. The function signal generator provided a sinusoidal periodic signal to the exciter, which drove the exciter to apply a sinusoidal periodic force to the folded energy harvester. It was also forced to vibrate. The waveforms of V*p*_1_, V*p*_2_, V*p*_3_ and V*p*_4_ were monitored in real time by a four-channel oscilloscope (GDS-1104B). The function signal generator started from 90 Hz to 140 Hz. The phase difference among V*p*_1_, V*p*_2_, V*p*_3_ and V*p*_4_ was recorded at every 1 Hz increase. With V*p*_1_ as the reference phase of 0°, the phase difference between the peak-to-peak value of V*p*_2_ and V*p*_1_, V*p*_3_ and V*p*_1_, and V*p*_4_ and V*p*_1_ were φ_2_, φ_3_ and φ_4_, respectively. φ_3_ was kept within 40°. The phases of V*p*_2_ and V*p*_4_ were essentially consistent. In the frequency range of 110–120 Hz, the phase of V*p*_2_ lagged at approximately 170° compared to V*p*_1_, and the phase of V*p*_4_ was about 170° ahead of V*p*_1_. In the frequency range of 95–105 Hz, the phases of the V*p*_1_, V*p*_2_, V*p*_3_ and V*p*_4_ peak-to-peak values tended to be consistent, and the phase relationship is shown in Figure 6.

### 2.3. Energy Harvesting Circuit

Earlier research on broadband energy harvesting circuits focused on the matching array of piezoelectric energy harvesters that combined cantilever beams with different resonance frequencies to achieve the purpose of broadband energy harvesting. However, this approach undoubtedly increased the volume of the piezoelectric harvesting energy, which was adverse for the miniaturization of the energy harvesting system. Instead, our proposed folded piezoelectric energy harvester led to four output terminals in a thin beam. The output characteristics of the four outputs at different frequencies also varied. Therefore, the miniaturization was achieved while achieving broadband energy harvesting in the folded structure piezoelectric energy harvester.

The basic topology of the synchronous electronic charge extraction circuit (SECE) based on the Buck structure is shown in Figure 7. The SECE was mainly composed of two diodes (*D*_1_, *D*_2_), two switches (*S*_1_, *S*_2_), an inductance (*L*) and an energy storage filter capacitor (*C_S_*), which greatly reduced the number of electronic components. For most of the time during a vibration cycle, switches *S*_1_ and *S*_2_ were in the off state, and the current source charged the internal capacitor *C_p_* [18].

When the open circuit voltage *V_p_* of the piezoelectric energy harvester reached the peak value, the energy accumulation in the piezoelectric material reached the maximum value. At this time, the switch was turned on and the energy accumulated in the internal equivalent capacitance *C_p_* was transferred to the inductance through the LC resonance. Taking the positive half cycle as an example, when the voltage *V_p_* was greater than zero, the switch *S*_1_ was turned on and the internal equivalent capacitor *C_p_*, the inductance *L* and the energy storage capacitor *C_S_* formed a CLC oscillating circuit. The charge in the capacitor *C_p_* was gradually released during the transfer process until the charge was completely transferred. At this stage, the energy transferred to the load was expressed as:(1)$$E_{1}=C_{p}V_{\mathrm{OC},\max}V_{\mathrm{DC}}$$
where *V*_OC,max_ is the maximum value of the open circuit voltage of the piezoelectric material. When the open circuit voltage of the PZT dropped to zero, the switch S_1_ was disconnected and the circuit entered into the inductance freewheeling phase, thereby forming a L-Cs-D_1_-D_2_ loop. At this time, the remaining energy in the inductance was transferred to the load. Assuming that the power loss of the circuit at this stage was not considered, the energy transferred to the load was expressed as:(2)$$E_{2}=\frac{1}{2}C_{p}V_{\mathrm{OC},\max}^{2}-E_{1}=C_{p}V_{\mathrm{OC},\max}\left(\frac{1}{2}V_{\mathrm{OC},\max}-V_{\mathrm{DC}}\right)$$

Through these two stages, the energy accumulated in the positive half cycle, as well as in the negative half cycle, was similarly transferred to the load.

The circuit was also suitable for harvesting the energy of multiple sources simultaneously. Given that the switching time was very short, the oscillation time of the equivalent capacitance and inductance in each charge source was also short. The charge transfer to the inductance was also completed instantly. Therefore, multiple sources shared an inductance L simultaneously, thereby reducing the volume of the energy harvesting system. In this paper, the piezoelectric vibrator with folded structure had four outputs, namely, V*p*_1_, V*p*_2_, V*p*_3_ and V*p*_4_. Through the phase analysis in Section 2.2, the phases of V*p*_1_ and V*p_3_* and V*p*_2_ and V*p*_4_ are essentially the same. Therefore, we adopted the scheme of harvesting V*p*_1_ and V*p*_3_ and V*p*_2_ and V*p*_4_ simultaneously. Figure 8 shows the multi-charge source input synchronous electronic charge extraction (MSI-SECE) circuit.

## 3. Experiments and Discussion

The experimental platform is shown in Figure 9. The folded piezoelectric energy harvester was installed on the exciter (SINOCERA PIEZOTRONICS JZK-40). The sine signal output by the function signal generator (RIGOL DG1022U) was driven by the power amplifier (SINOCERA-YE5874A) to simulate the vibration environment of the tractor. The acceleration sensor (LC0408T) monitored the acceleration of the vibration platform in real time. The charge amplifier (DLF-4) and the data acquisition instrument (m+p VibPilot) transmitted the acceleration signal to the host computer in real time. At the same time, the MSI-SECE circuit was used as the vibration energy acquisition circuit, and its component parameters are shown in Table 4.

Based on the phase output relationship of the folded piezoelectric energy harvester described in Section 2.2, the energy output from V*p*_1_ and V*p*_3_ and V*p*_2_ and V*p*_4_ of the MSI-SECE circuit was extracted simultaneously. The harvested energy was used separately to avoid the energy reflux caused by the phase relationship. Figure 10 shows the voltage waveform of the MSI-SECE circuit when harvesting the output energy of V*p*_1_ and V*p*_3_ simultaneously. The phase difference source voltage waveform was 9.9°. The small phase difference demonstrated that the harvesting policy is desirable.

Figure 11 describes the power output through the sweeping frequency of MSI-SECE circuit at accelerations of 0.5, 1 and 1.5 m/s^2^. Figure 11a shows that the energy output from V*p*_1_ and V*p*_3_ is harvested simultaneously. When the constant acceleration was 1.5 m/s^2^, the average energy harvesting power in the vibration frequency range of 90–140 Hz was 321.8 μW and the maximum harvesting power at 121.5 Hz was 592.9 μW. When the constant acceleration was 0.5 m/s^2^, the average normalized power density was 6.48 μW/(mm^3^·g^2^) (equal to the power divided by (the volume multiplied by the square of the acceleration)) in the range of 90–140 Hz and 11.65 μW/(mm^3^·g^2^) in the range of 122–123.5 Hz. Figure 11b shows that the energy output from V*p*_2_ and V*p*_4_ was harvested simultaneously. When the constant acceleration was 1.5 m/s^2^, the average energy harvesting power in the vibration frequency range of 90–140 Hz was 181.9 μW and the maximum harvesting power was 511 μW at 122 Hz. When the constant acceleration was 0.5 m/s^2^, the average normalized power density in the range of 90–140 Hz and 119–120 Hz was 3.40 μW/(mm^3^·g^2^) and 12.40 μW/(mm^3^·g^2^), respectively. In the experiment of simulating the vibration process of the tractor, the MSI-SECE circuit combined with the folded piezoelectric vibrator was harvested efficiently.

We also aimed to emphasize the effectiveness of using the folded piezoelectric energy harvester and MSI-SECE circuits compared to other energy harvesters. Table 5 shows the average normalized power density of the operable bandwidth of other devices. Notably, in the calculation process, our equipment considered the space occupied by the whole energy harvester rather than the volume of the material. Our folded piezoelectric energy harvester provided a considerable bandwidth and generated more energy per unit volume.

To harvest the vibration energy of the tractor exhaust tube to supply power to the sensor equipment, this paper selected the high-precision atmospheric pressure sensor MS5607, which was widely used in various fields as the power supply experiment object. In Figure 12, MS5607 is a new generation of high-speed resolution altimeter sensor with SPI and I^2^C interfaces. The sensor module included a high linear pressure sensor and an ultra-low power 24-bit Σ-ΔADC, which almost matches most microcontrollers on the market. It has a simple communication protocol and the internal registers in the device do not need to be programmed. Its volume is 5 mm × 3 mm × 1 mm.

The VDD and GND pins of the pressure sensor MS5607 were connected to the load end of the MSI-SECEE circuit. The resistance in parallel with 100 kΩ was used as the matching resistance. The output signal of the SDO pin was displayed by an oscilloscope in real time. By observing the duty ratio of the square wave signal output by the SDO pin of the pressure sensor, we judged whether the energy output by the MSI-SECEE circuit can cause the pressure sensor to work properly in the vibration environment. As shown in Figure 13a, the pressure sensor output a square wave signal with a stable duty cycle when the ambient pressure was stable. In the experiment, the ambient pressure was changed artificially (Figure 13b). The duty cycle of the square wave output by SDO changed and the middle vacancy part corresponded to the signal change after the pressure changes.

Table 6 records the output state of the SDO pin of the MS5607 pressure sensor under the vibration frequency range of 90–140 Hz and the acceleration of 0.5–1.5 m/s^2^ (*a* = 0.5–1.5 m/s^2^), respectively. V*p*_1_ and V*p*_3_ were used as inputs to drive the sensor harvested simultaneously by the MSI-SECE circuit with a greater effect. With the increase in acceleration, the effective working frequency band was gradually widened. When the frequency was 90–93 Hz, V*p*_1_ and V*p*_3_ as the input power did not drive the sensor to operate, but V*p*_2_ and V*p*_4_ met the energy demand. When the frequency was 125–140 Hz, V*p*_2_ and V*p*_4_ as the input power did not drive the sensor, but V*p*_1_ and V*p*_3_ met the energy demand. Obviously, the results show that the four charge sources are divided into two groups that produce complementary frequency ranges and ensure that the pressure sensor can output the signal normally in the frequency range of 90–140 Hz.

## 4. Conclusions

In this study, a broadband vibration energy harvester that can be applied to the vibration of the tractor exhaust cylinder was designed and developed. We analyzed the PSD of the measured data of the three directions, five measurement points and five constant rotational speeds of the tractor exhaust in the running state. In the frequency range of 20–390 Hz, the overall vibration of the exhaust was relatively intense. Meanwhile, the working frequency band of the piezoelectric vibrator with the folded structure matched the vibration of the tractor exhaust cylinder well to improve the energy harvesting efficiency. Finally, the experimental platform was used to simulate the vibration environment of the tractor during operation. The maximum normalized power density reached 12.40 μW/(mm^3^·g^2^) when the folded energy harvester was combined with the MSI-SECE circuit. In the frequency range of 90–140 Hz, the average normalized power density was 6.48 μW/(mm^3^·g^2^), in which the MS-5607 pressure sensor worked normally.

## Figures and Tables

**Figure 1 sensors-23-00286-f001:**
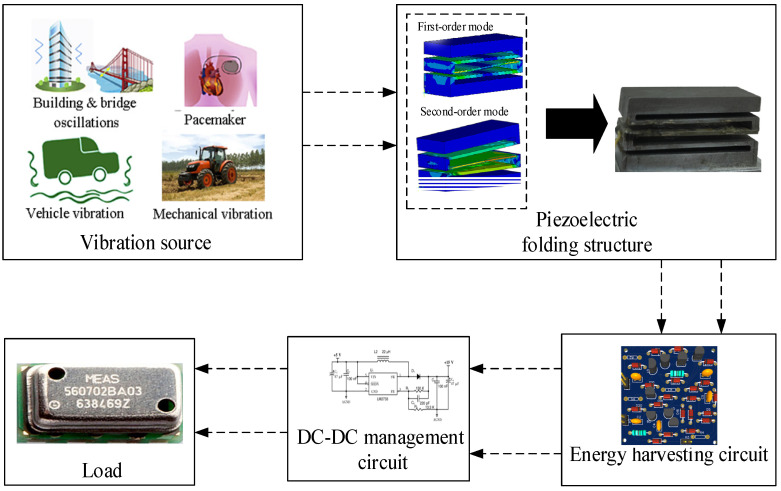
Piezoelectric energy harvesting system.

**Figure 2 sensors-23-00286-f002:**
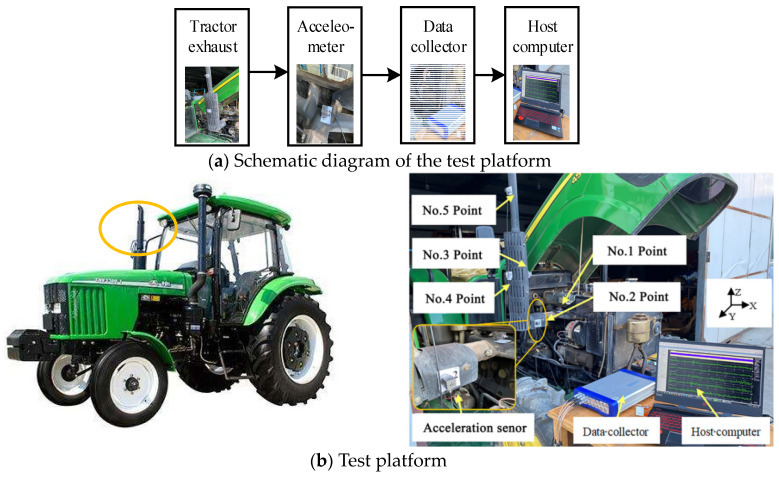
Vibration test.

**Figure 3 sensors-23-00286-f003:**
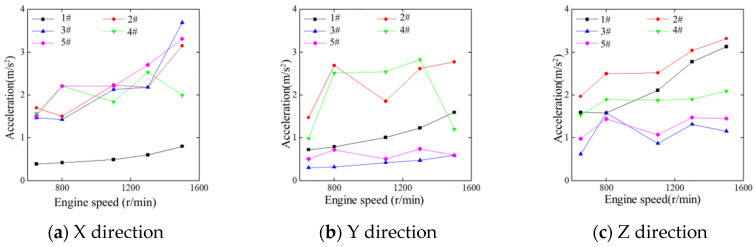
Comparison of the time domain analysis results of the exhaust cylinder vibration,where 1# is the NO.1point in Table 1. And so on.

**Figure 4 sensors-23-00286-f004:**
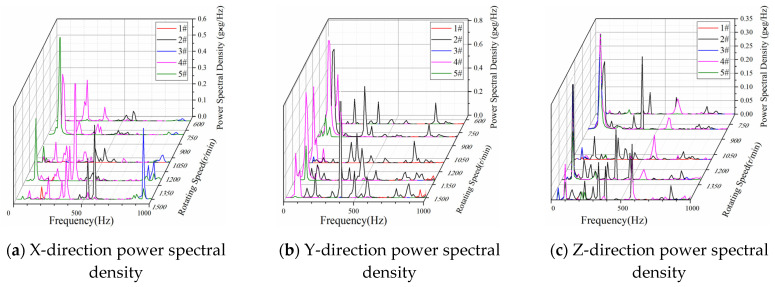
Frequency Domain Analysis of the Exhaust Tube Vibration.

**Figure 5 sensors-23-00286-f005:**
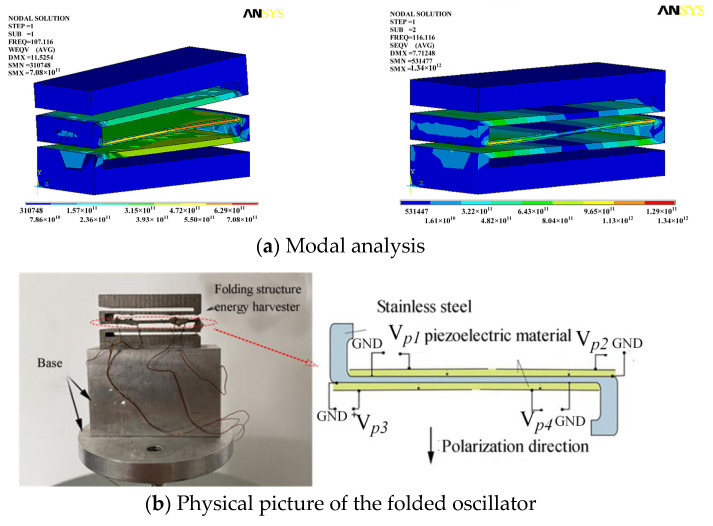
Folded structure piezoelectric energy harvester.

**Figure 6 sensors-23-00286-f006:**
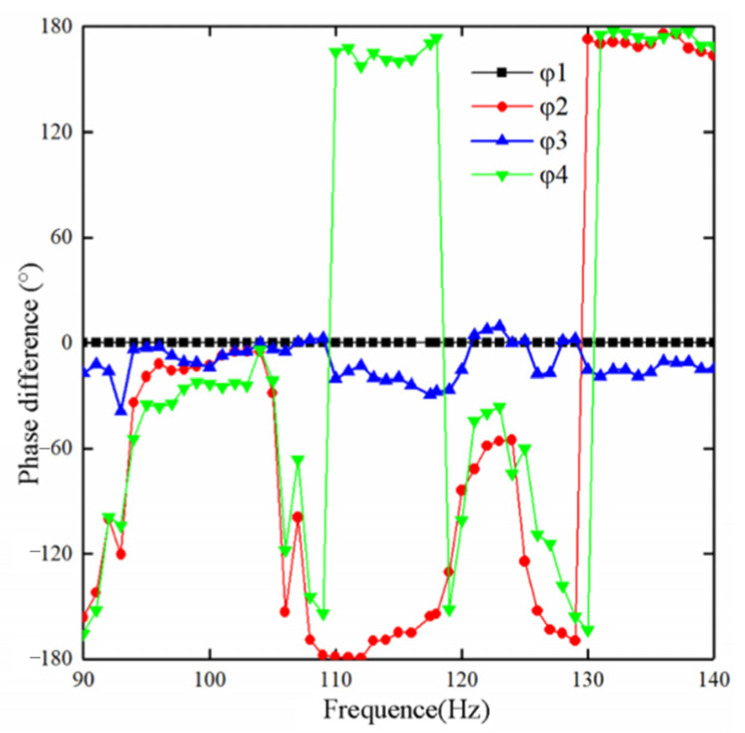
Voltage phase relationship of the four outputs.

**Figure 7 sensors-23-00286-f007:**
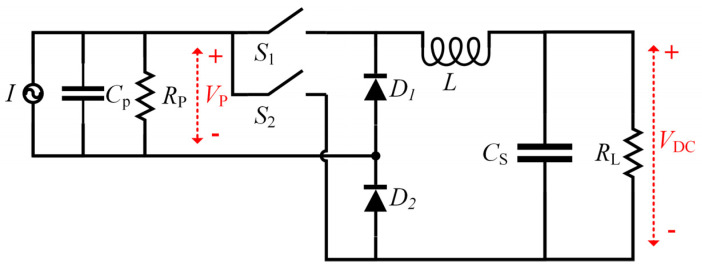
The SECE based on the Buck structure.

**Figure 8 sensors-23-00286-f008:**
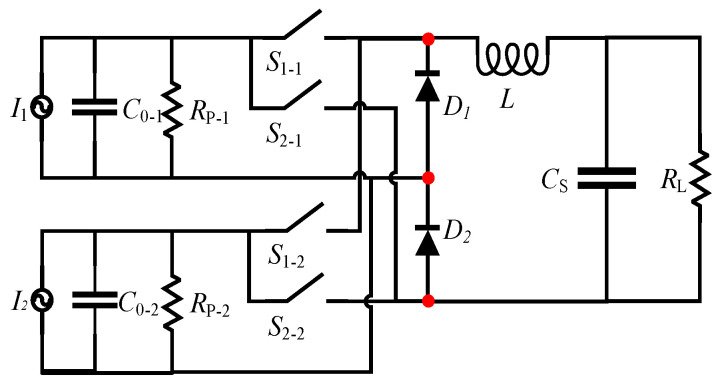
MSI-SECE.

**Figure 9 sensors-23-00286-f009:**
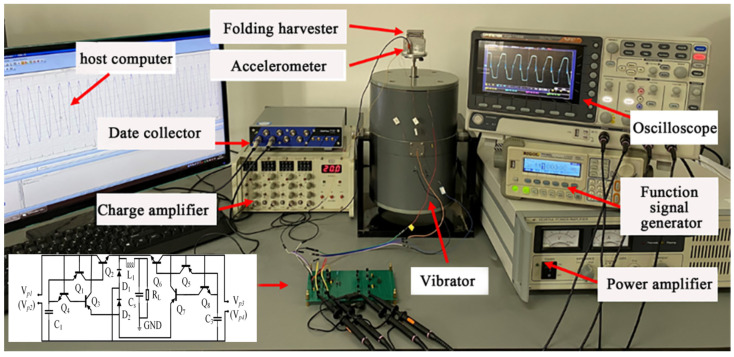
Experimental platform.

**Figure 10 sensors-23-00286-f010:**
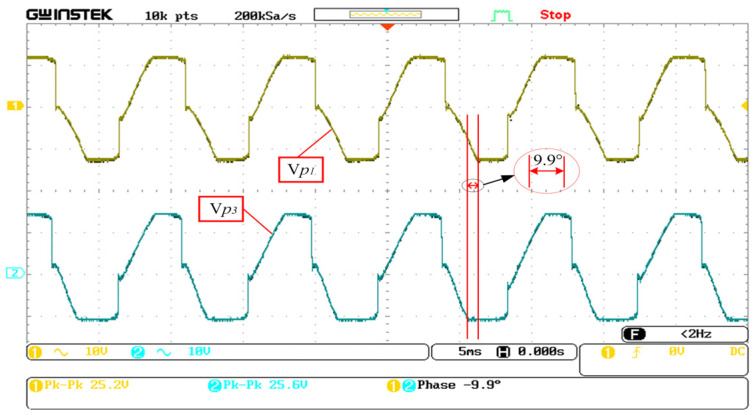
Phase difference between V*p*_1_ and V*p*_3_.

**Figure 11 sensors-23-00286-f011:**
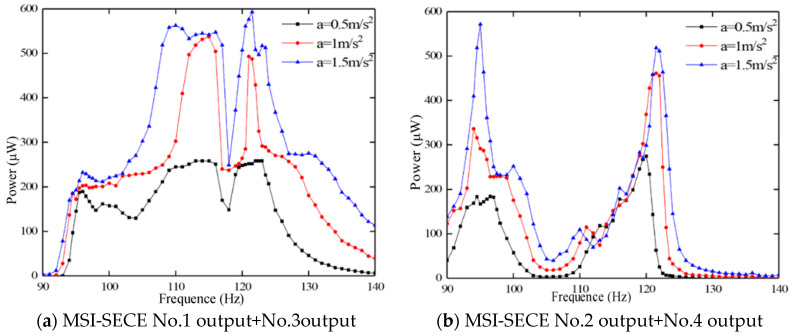
The sweep power output of the MSI-SECE circuit.

**Figure 12 sensors-23-00286-f012:**
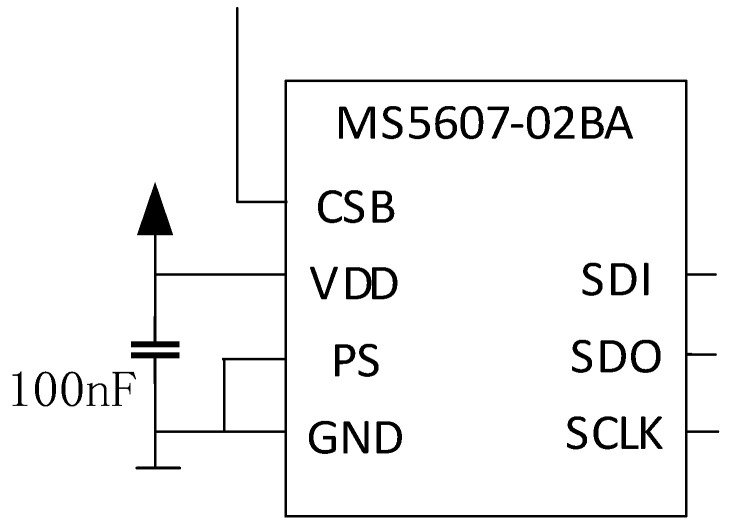
Typical application circuit of MS5607 barometric pressure sensor.

**Figure 13 sensors-23-00286-f013:**
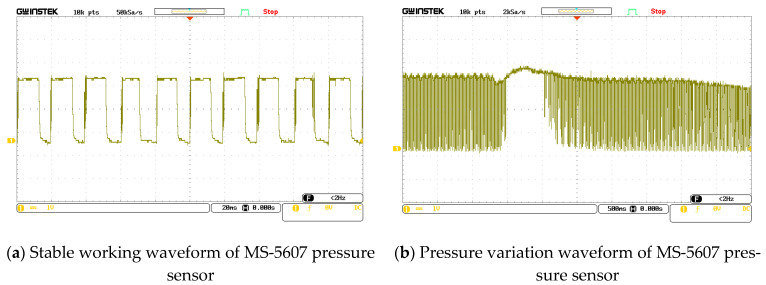
Working waveform of the barometric pressure sensor when the power was supplied by vibration energy.

**Table 1 sensors-23-00286-t001:** Arrangement of measuring points for the tractor exhaust cylinder.

Exhaust Cylinder Measuring Point	
No. 1 point	Upper part of exhaust cylinder
No. 2 point	Exhaust cylinder muffler front side
No. 3 point	Exhaust cylinder muffler right
No. 4 point	Exhaust cylinder muffler left
No. 5 point	Left side of tail cylinder

**Table 2 sensors-23-00286-t002:** INV3062S data acquisition instrument.

Main Specifications	
Number of channels	8
Sampling frequency	6.25–102.4 kHz
Analysis broadband	50 kHz
Input coupling	ICP
Input resistance	>1 MΩ

**Table 3 sensors-23-00286-t003:** Structural parameters of piezoelectric materials.

Material	Material Parameters	Numerical Value	Material Parameters	Numerical Value
p-51	*S^E^*_11_/×10^−12^ m^2^·N^−1^	15.0	*d*_31_/×10^−12^ C·N^−1^	−185
	*S^E^*_12_/×10^−12^ m^2^·N^−1^	−4.8	*d*_33_/×10^−12^ C·N^−1^	400
	*S^E^*_13_/×10^−12^ m^2^·N^−1^	−6.75	*D*_15_/×10^−12^ C·N^−1^	650
	*S^E^*_33_/×10^−12^ m^2^·N^−1^	17.6	*ε^T^*_33_/*ε*_0_	2100
	*S^E^*_44_/×10^−12^ m^2^·N^−1^	41.5	*ε^T^*_11_/*ε*_0_	2400
	*S^E^*_66_/×10^−12^ m^2^·N^−1^	39.6	*ε^s^*_33_/*ε*_0_	1071
	*S^D^*_11_/×10^−12^ m^2^·N^−1^	13.1	*ε^s^*_11_/*ε*_0_	1290
	*S^D^*_12_/×10^−12^ m^2^·N^−1^	−6.7	*p*/×10^3^ kg·m^−3^	7.45
	*S^E^*_13_/×10^−12^ m^2^·N^−1^	−2.78	Q_m_	100
	*S^D^*_33_/×10^−12^ m^2^·N^−1^	9.0	*k* _31_	0.36
	*S^D^*_44_/×10^−12^ m^2^·N^−1^	22.0	*tgδ%*	2.0
Stainless steel 304	*E*/GPa	200	µ	0.28
	ρ/kg×m^−3^	7930	-	-

**Table 4 sensors-23-00286-t004:** List of components.

Components	Parameter
NPN	2N3904
PNP	2N3906
*L*	10 mH
D_1_-D_2_	1N4148
*C* _1_	1 nF
*C* _S_	10 μF
*R* _L_	100 kΩ

**Table 5 sensors-23-00286-t005:** Comparison of the contemporary VEH devices with this work.

Reference	Size	Bandwidth/Operable Frequency (Hz)	Power Density (μW/mm^3^g^2^)
Alghisi, D [19]	45 mm × 19 mm × 1.8 mm	12 Hz at 0.5 g	0.93 at 42.3 Hz
Yucheng [20]	85 mm × 20 mm × 1 mm, 83 mm × 20 mm × 1 mm, 80 mm × 20 mm × 1 mm	8 Hz at 2 m/s^2^	0.35 at 59.5 Hz
Kp A [21]	35 mm × 19 mm × 0.25 mm	9 Hz at 0.5 g 11 Hz at 1 g	2.66 at 78 Hz
Challa, V. R. [22]	34 mm × 20 mm × 0.6 mm	22–32 Hz at 0.8 m/s^2^	0.88 around 22–32 Hz
Kp. A [21]	38 mm × 26.3 mm × 0.25 mm	29 Hz at 0.5 g 45 Hz at 1 g	1.69 at 106 Hz
Song, H. C. [23]	5 mm × 50 mm × 0.35 mm	30 Hz at 1 g	0.24 at 78 Hz
Li X Y [24]	110 mm × 22 mm × 1.5 mm	17 Hz 0.02 g	7.69 at 27.36 Hz
This work	34 mm × 13.6 mm × 18.4 mm	50 Hz at 0.5 m/s^2^	12.39 at 120 Hz 6.48 around 90–140 Hz

**Table 6 sensors-23-00286-t006:** Operating status of the air pressure sensor at a 90–140 Hz vibration frequency.

Sensor Working Condition (m/s^2^)	V*p*_1_ and V*p*_3_ as Input			V*p*_2_ and V*p*_4_ as Input		
	*a* = 0.5	*a* = 1	*a* = 1.5	*a* = 0.5	*a* = 1	*a* = 1.5
Operable frequency (Hz)	93–132	93–140	92–140	90–102 110–124	90–125	90–128

## Data Availability

The data that support the findings of this study are available from the corresponding author upon reasonable request.

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
