# Peer review of "Design and Development of a Broadband Vibration Energy Harvester Suitable for Tractor Exhaust Cylinder Vibration"

_sensors, 2022, doi:10.3390/s23010286_

Round 1

Reviewer 1 Report

This work explored the vibration of John Deere-454 exhaust cylinder, harvested the vibration energy using piezoelectric vibration energy harvesting technology, and drove the pressure sensor for tractor exhaust cylinder to work normally in a wide frequency range. The subject is interesting and meaningful. The specific comments are as follows:

1. Reconsider the title of Fig. 1

2. The base part in Fig.5(b) seems large and heavy, does the base part have particular usage? What is the impact of the base part on your VEH system?

3. Fig. 12(a) may have a copyright issue, take care of this.

4. Fig. 5(a) and line 174. The modal simulation analysis of the folded energy harvester is carried out by ANSYS, does the author own the copyright of ANSYS? Or the simulation was made by someone else who has a legal copyright?

5. Fig. 5(b), I don’t think “Physical figure” is a proper title.

6. Fig. 5(c) should contains texts of “vp1”, “vp2”, “vp3” and “vp4”, referring to line 189 and 190.

7. Line 187, what does the “ZT” mean?

8. Fig. 9 has a blank textbox, check this.   

Reviewer 2 Report

The author explored the vibrations generated during tractor operation and bumping, and tried to convert the vibration energy into electrical energy through piezoelectric means for collection and utilization. A high-efficiency broadband piezoelectric vibration energy harvesting system and its application scheme are proposed in this paper. The manuscript is not ready for publication and requires a minor revision. The specific comments are listed as below:

(1)   The introduction part can be improved by introducing more references of vibration energy harvesting, such as multi-modal vibration energy harvester.

(2)   In figure 9, there exists a white square domain in the figure, please delete it.

(3)   Could the MOSFET can be used instead of Q1 and Q2 to make the circuit loss lower?

Reviewer 3 Report

The authors propose obtaining the vibration energy of the tractor exhaust and converting it into electricity. This problem is not new and has been addressed by various research groups. The efforts of the authors in the experimental research carried out, as well as the elaboration of their results, should be appreciated.
I have a few doubts and I would like to ask the authors to respond to them:

1. The authors refer to the results of previous studies, but do not provide a reference to them, nor do they describe exactly what was done in previous studies and how they differ from the presented ones. Please extend the reference and add a link to the previous publication so that the reader can compare the results.

2. Figure 4 - probably the same scale in all figures would give a better comparison.

3. Figure 9 - there is a white rectangle in the lower left corner. Please check.

4. Some minor typo or unfinished sentences, for example:

line 284 - " divided by ...".

5. Figure 11 - probably the most important graph, which shows that configuration No. 1 allows you to obtain the most power in the widest range, while configuration No. 2 is the same as a linear system with two characteristic frequencies. I have two questions:

a. why in the case of a. there is a clear power drop around 118 Hz?

b. why in case b. the system behaves in a linear way?

6. Please check the labels on all figures all and combine many labels to single label.

Round 2

Reviewer 3 Report

I would like to thank the authors for addressing all my comments and providing exhaustive answers to my questions.